# Integrated Metabolomics Approach Reveals the Dynamic Variations of Metabolites and Bioactivities in *Paeonia ostii* ‘Feng Dan’ Leaves during Development

**DOI:** 10.3390/ijms25021059

**Published:** 2024-01-15

**Authors:** Zhangzhen Bai, Junman Tang, Yajie Li, Zhuoning Li, Siyi Gu, Lu Deng, Yanlong Zhang

**Affiliations:** 1College of Landscape Architecture and Arts, Northwest A&F University, Yangling 712100, China; baizhangzhen@nwafu.edu.cn (Z.B.); tangjunman@nwafu.edu.cn (J.T.); liyajie@nwafu.edu.cn (Y.L.); lizhuoning@nwafu.edu.cn (Z.L.); gusiyi@nwafu.edu.cn (S.G.); 2College of Animal Science and Technology, Northwest A&F University, Yangling 712100, China

**Keywords:** *Paeonia ostii*, leaf development, integrated metabolomics approach, antioxidant, antibacterial property

## Abstract

*Paeonia ostii* ‘Feng Dan’ is widely cultivated in China for its ornamental, medicinal, and edible properties. The whole plant of tree peony is rich in bioactive substances, while the comprehensive understanding of metabolites in the leaves is limited. In this study, an untargeted metabolomics strategy based on UPLC-ESI-TOF-MS was conducted to analyze the dynamic variations of bioactive metabolites in *P. ostii* ‘Feng Dan’ leaves during development. A total of 321 metabolites were rapidly annotated based on the GNPS platform, in-house database, and publications. To accurately quantify the selected metabolites, a targeted method of HPLC-ESI-QQQ-MS was used. Albiflorin, paeoniflorin, pentagalloylglucose, luteolin 7-glucoside, and benzoylpaeoniflorin were recognized as the dominant bioactive compounds with significant content variations during leaf development. Metabolite variations during the development of *P. ostii* ‘Feng Dan’ leaves are greatly attributed to the variations in antioxidant activities. Among all tested bacteria, the leaf extract exhibited exceptional inhibitory effects against *Streptococcus hemolytis*-*β*. This research firstly provides new insights into tree peony leaves during development. The stages of S1–S2 may be the most promising harvesting time for potential use in food or pharmaceutical purposes.

## 1. Introduction

Tree peony (*Paeonia* Section *Moutan*) is indigenous to China and belongs to the family Paeoniaceae [1]. It has been used for ornamental and medicinal purposes for thousands of years. In recent years, the planting area of tree peony rapidly expanded, potentially reaching 200,000 hectares [2], following the discovery of health-promoting α-linolenic acid in its seeds [3,4,5].

To adapt to changing environments and various stresses, plants need to biosynthesize a large variety of metabolites [6]. From another perspective, the plant kingdom serves as a natural resource base for health-beneficial metabolites. *Paeonia* species, as medicinal plants, are abundant in natural functional ingredients. Over 451 metabolites have been isolated and identified in peonies, falling under various clades, such as flavonoids, tannins, terpenoids, stilbenes, and phenols [7,8]. The root bark of the tree peony, traditionally used in Chinese pharmacopeia, is considered to be medicinal and possesses a range of pharmaceutical properties including antioxidant, antibacterial, anti-tumor, anti-inflammatory, anti-diabetes, and anti-osteoporosis, among others [9,10,11]. The metabolites of its root bark have undergone comprehensive investigation, identifying paeoniflorin, paeonol, 1,2,3,4,6-penta-O-galloylglucose, methyl gallate, and gallic acid as the targeted active ingredients [12]. Additionally, the phytochemical composition and biological activities of peonies, including seeds, flowers, and stems, have been extensively studied. These investigations illustrate that the extracts from the tree peony exhibit antioxidant, antibacterial, anti-inflammatory, anti-photoaging, and antidiabetic activities due to the presence of monoterpene glycosides, flavonoids, stilbenes, tannins, and other bioactive compounds [13,14,15,16]. However, a comprehensive understanding of metabolites in tree peony leaves has yet to be fully demonstrated.

Tree peony leaves (TPLs), a by-product of this widely cultivated economic crop, hold urgent potential for development and utilization. In traditional folk medicine, TPLs have been used to treat chronic cervicitis and infantile abdominal pain. The *History of Bozhou Tree Peony* in the Ming Dynasty mentions that the leaf buds of the tree peony can be processed into tea with a special and lasting fragrance. Recent studies have identified over 20 metabolites in TPL, including monoterpene glycosides, tannins, phenols, and flavonoids [15]. Moreover, Bai et al., (2022) found that TPLs contain more abundant gallotannins than other parts of tree peonies, both in terms of content and molecular structure [17]. Additionally, Zhang et al., (2019) observed a significant variation in paeoniflorin during TPL development, with the highest amount occurring at the leaf bud stage [18]. However, the current research only covers a limited number of compounds, hindering a comprehensive understanding of TPL metabolites.

LC-MS-based metabolomics has emerged as a highly efficient and indispensable tool for the high-throughput characterization of metabolites [19]. In the present study, *Paeonia ostii* ‘Feng Dan’ was used as plant material due to its wide cultivation in China. Untargeted and targeted metabolomics approaches were first conducted to comprehensively determine the metabolites present in TPLs during their development. Furthermore, potential bioactivities, including antioxidant and antibacterial activities, were dynamically evaluated. The findings of this work will serve as a theoretical foundation for the comprehensive utilization and exploitation of *P. ostii* ‘Feng Dan’ leaves.

## 2. Results and Discussion

### 2.1. Tree Peony Leaves Are Rich in Phenols and Flavonoids

Plant secondary metabolites play a crucial role in responding to biotic and abiotic stresses [20]. Among these metabolites, polyphenols are widely distributed throughout the plant kingdom and have multiple beneficial effects on human health [21]. To examine the variations in polyphenol content in tree peony leaves during growth and development, the total phenolic content (TPC) and total flavonoid content (TFC) were investigated. As illustrated in Figure 1, the TPC of *P. ostii* ‘Feng Dan’ initially decreased, then increased, and eventually stabilized. The highest TPC value was 10.21 g GE/100 g DW at the S11 stage, while the lowest value was 7.09 g GE/100 g DW at the S3 stage. During the stabilized period, the TPC remained around 9 g GE/100 g DW. In contrast, the variation in TFC of *P. ostii* ‘Feng Dan’ showed an opposite trend. The highest TFC value, accounting for 1.01 g RE/100 g DW, was observed at the S3 stage, whereas the lowest value was 0.25 g RE/100 g DW at the S8 stage. The initial and final values were 0.45 g RE/100 g DW (S1) and 0.52 g RE/100 g DW (S17), respectively.

The accumulation of secondary metabolites in plants could be influenced by many factors, including temperature, light, soil properties, water, and chemical stress [22]. It was observed that the TPC and TFC of TPL suddenly dropped at the stage of S8, which may be possibly explained by the light. Because there is more precipitation in May, consecutive rainy and cloudy days are mainly distributed between S7 and S8 (Appendix A). Sunlight, especially UV-B, is one of the important factors influencing the biosynthesis of polyphenols [23,24]. Insufficient light led to sudden decreases of TPC and TFC in TPL at S8. The accumulation of secondary metabolites may also be affected by development [25]. It is worth noting that both the TPC and TFC variations of tree peony leaves exhibited a distinct turning point at stage S3. It happened to be the transition of peonies from vegetative to reproductive growth (Appendix A, Appendix A), indicating this transition may be the key factor that influenced the TPCs and TFCs during TPL development. Considering the influence of environmental factors and development, an investigation of their variations during TPL development still needs to be conducted in the next growing season.

Although the TPC varied through all development stages of tree peony leaves, the TPC of tree peony leaves was significantly higher at most stages compared to other parts of the tree peony, such as roots, stems, flowers, and seeds [13,15,26,27]. This indicates that leaves contain the highest concentration of polyphenols among all tree peony tissues. Consequently, tree peony leaves can serve as an important natural source of polyphenols with significant development potential.

### 2.2. Annotation of Metabolites through Non-Targeted Metabolome Approach

To comprehensively investigate the metabolic composition and variations in TPL during development, a non-targeted metabolome strategy based on LC-QTOF-MS was employed. This approach offers the advantages of high throughput and high resolution [28]. The data for TPL across all stages were acquired using the Data-Dependent Acquisition (DDA) mode. The annotation of TPL metabolites in this matrix was conducted based on the Global Natural Products Social Molecular Networking (GNPS) platform [29], an in-house MS database, and published literature. Ultimately, a total of 321 compounds were annotated, with the majority classified as flavonoids, polyphenols, and terpenoids (Appendix A).

### 2.3. Multivariate Statistical Analysis of Non-Targeted Metabolome Data

We processed the raw data of each sample using MZmine 2 software. After peak detection, grouping of isotopic peaks, and peak alignment, a data matrix 1353 × 54 in size (consisting of 17 stages and one quality control of TPL in triplicate) was obtained. To analyze the variation pattern of TPL metabolites, the matrix was imported into SIMCA 14.1 software to establish a principal component analysis (PCA) model. As shown in Figure 2a, all quality controls (QCs) in biological triplicate were successfully clustered together, confirming the good repeatability of the non-targeted metabolome approach. The sum of the first and second principal components accounts for 57.3% of the total variance, capturing most of the variable information. The scatter plot of PCA illustrates a clear separation between the period from S1 to S7 (Group 1) and the period from S8 to S17 (Group 2). Additionally, the hierarchical clustering analysis (HCA) also exhibits a distinct separation of the two groups (Figure 2b). This finding is consistent with the results of TPC and TFC, indicating a significant difference in the content between Group 1 and Group 2. This phenomenon may be mainly affected by the transition of peonies from vegetative to reproductive growth.

Additionally, partial least-squares discriminant analysis (OPLS-DA) was used to construct a model for the two groups. A random permutations test was conducted, revealing R2 and Q2 values of 99.72% and 78.98%, respectively, indicating the high quality and reliability of the OPLS-DA model. Additionally, the score plot for the OPLS-DA model (Figure 2c) clearly shows the separation of individuals from the two groups, which can be attributed to the presence of differential metabolites. The S-plot visually represents the differential metabolites (marked in red) with VIP values greater than 1 (Figure 2d). By applying a threshold of fold change (FC ≥ 2 or FC ≤ 0.5), a total of 602 differential compounds were observed between Group 1 and Group 2. Among these, only 117 compounds were known metabolites (Figure 3). Cluster heatmap analysis reveals that the first seven stages of leaves cluster together, while the last ten stages cluster separately, suggesting that these known metabolites are important markers for distinguishing Group 1 and Group 2. Notably, the early stages of TPL exhibited higher contents of albiflorin, oxypaeoniflorin, paeoniflorin, rutin, and tetragalloylglucose, whereas the late stages of TPL had higher contents of Asiatic acid, olivil 4′-O-glucoside and salidroside (Figure 3).

During the development of TPL, there was a significant variation in differential metabolites. To identify the metabolic pathways related to the differential metabolites in Group 1 and Group 2, a KEGG enrichment analysis was conducted (Figure 4). This analysis revealed that these differential metabolites were predominantly enriched in 17 metabolic pathways. Of particular importance were the pathways involved in the biosynthesis of flavone, flavonol, and phenylpropanoids, as they contained a substantial number of differential metabolites. Furthermore, monoterpenoid biosynthesis emerges as a noteworthy pathway, with several monoterpenoids identified among differential metabolites, including paeoniflorin, albiflorin, and oxypaeoniflorin. These monoterpene glycosides were characteristic metabolites of Paeoniaceae, although the biosynthesis pathway remains unclear. While the specific pathway for the biosynthesis of these monoterpene glycosides could not be fully illustrated in Figure 4, the high content and significant variation observed in the TPL during development could serve as a potential clue for unraveling the biosynthesis pathway of monoterpene glycosides in the TPL system.

### 2.4. Quantification of Metabolites through Targeted LC-QQQ-MS Analysis

The LC-triple-TOF-MS-based full-scan MS method offers the advantage of high resolution for facilitating metabolite identification. However, the detector can be easily saturated, thereby limiting its linear range and hindering the accurate quantification of metabolites [30]. To address this limitation, a targeted analysis utilizing LC-QQQ-MS was performed to precisely monitor the variations of selected compounds during the development of TPL. A total of twenty-two compounds, comprising thirteen flavonoids, one tannin, four terpenoids, and four other phenolic compounds, were dynamically quantified using previously established MRM methods [31].

To assess the precise composition of selected metabolites in TPL throughout its development, a heatmap displaying the targeted metabolites was generated. As illustrated in Figure 5a, the metabolites albiflorin, paeoniflorin, pentagalloylglucose (PGG), luteolin 7-glucoside, and benzoylpaeoniflorin were found to be the major constituents of TPL. The highest concentrations of these compounds were observed at the S4, S4, S1, S3, and S1 stages, amounting to 2669.33 mg/100 g DW, 1359.17 mg/100 g DW, 590.34 mg/100 g DW, 451.05 mg/100 g DW, and 108.20 mg/100 g DW, respectively (Appendix A).

Although Figure 5a provides a visualization of the major metabolites in TPL, it cannot illustrate the variations in each target metabolite at different development stages. Therefore, another heatmap was plotted with each target metabolite in a row independently (Figure 5b). Obviously, the highest contents of most metabolites occurred during the initial stages and then gradually or rapidly decreased at the S3 stage. On the other hand, the contents of certain metabolites, such as methyl gallate and hyperoside, increased during the final stages of TPL development. These observations partly explained the separation of TPLs at different stages, as depicted in Figure 2.

Overall, the predominant compounds in leaves of *P. ostii* ‘Feng Dan’ were albiflorin, paeoniflorin, PGG, luteolin 7-glucoside, and benzoylpaeoniflorin, with the highest contents exceeding 100 mg/100 g DW. These metabolites exhibited significant fluctuations during the development of TPL. The turning point occurred around the stage of S4, the blooming period of the tree peony. However, after flowering, the levels of these compounds decreased to a stable trend. It can be explained by the development, especially the transition of the peony from vegetative to reproductive growth. Albiflorin, paeoniflorin, and benzoylpaeoniflorin are characteristic metabolites of Paeoniaceae that possess various bioactivities, including anti-tumor, neuroprotective, antidepressant-like, and antimicrobial effects [32,33,34,35]. Paeoniflorin is the most typical monoterpenoid of the tree peony. Although a similar variation trend of paeoniflorin during development was also observed in a published work, the degree of decline varied greatly. The reasons may be tree age, environmental factors, etc. PGG also exhibits multiple health-promoting properties, including antioxidant, anticancer, antiviral, anti-inflammatory, and antidiabetic activities [36]. Therefore, these bioactive compounds have great potential for development and utilization in TPL. Our findings regarding the dynamic changes in these compounds provide valuable guidance for determining the optimal harvesting time.

### 2.5. Antioxidant Activities of TPLs at Different Stages

Although the antioxidant activities of *P. ostii* leaves at the bud stage, including ABTS^+•^, DPPH^•^, FRAP, and ORAC, have been investigated [37], their dynamic variation patterns during leaf development are still unclear. In general, bioactive constituents contribute to the biological activities of TPL. The antioxidant activities of TPL change dynamically due to the variations in metabolites during development. To monitor these changes, ABTS^+•^, DPPH^•^, FRAP, and ORAC assays were conducted. The result showed that the ABTS^+•^, DPPH^•^, FRAP, and ORAC activities of TPL at S1 are essentially consistent with the previous report [38]. A similar trend in ABTS^+•^-scavenging capacity, DPPH^•^-scavenging capacity, and FRAP was observed, which initially decreased, followed by an increase to a relatively stable state for certain stages, and finally showed a slight decrease (Figure 6). The highest and lowest activities of ABTS^+•^-scavenging capacity and FRAP occurred at the same stages. At stage S1, the strongest activities were observed, with ABTS^+•^-scavenging capacity accounting for 32.90 g TE/100 g DW and FRAP accounting for 27.00 g TE/100 g DW. On the other hand, at stage S4, the weakest activities were observed, with ABTS^+•^-scavenging capacity accounting for 17.53 g TE/100 g DW and FRAP accounting for 11.54 g TE/100 g DW. The DPPH^•^-scavenging capacity of TPL reached its highest and lowest values at stages S13 and S4, accounting for 49.34 g TE/100 g DW and 24.85 g TE/100 g DW, respectively. The strongest ORAC activity of TPL was observed at stage S1 with a value of 21.71 TE/100 g DW, while the weakest ORAC activity (6.64 g TE/100 g DW) was found at the last stage.

The antioxidant activity of TPL extracts is attributed to the presence of bioactive compounds. Pearson correlation analysis revealed a close relationship between the scavenging activities of TPL against ABTS^+•^ and DPPH^•^ radicals and the TPC (Figure 7, Appendix A). This finding emphasized the important role of polyphenols in scavenging ABTS^+•^ and DPPH^•^ radicals, which is consistent with previous studies [39,40]. Although weak correlations were observed between the activities (FRAP and ORAC) and the total constituents (TPC, TFC, and TMC), specific bioactive metabolites showed a stronger association with FRAP and ORAC activities. Notably, benzoic acid, gallic acid, benzoylpaeoniflorin, oxypaeoniflorin, and PGG, which are present in high contents in TPL, exhibited a close correlation (r > 0.4) and are likely to be the major bioactive compounds responsible for FRAP (Figure 7, Appendix A). Our analysis revealed a highly intricate network between ORAC and bioactive constituents. Benzoic acid, quercetin, (+)-catechin, isorhamnetin, kaempferol, and astragalin showed correlation coefficients higher than 0.5, although only the content of benzoic acid was relatively abundant (Figure 7, Appendix A). Therefore, the ORAC of TPL may be influenced by the interaction of these related metabolites.

### 2.6. Antibacterial Activities of TPLs at Different Stages

To investigate the antibacterial potential, the TPL extracts of seventeen different stages were evaluated using the minimum inhibitory concentration (MIC) method. A lower MIC value indicates a stronger inhibitory effect on bacterial growth. The results demonstrated that all TPL extracts exhibited a significant inhibitory effect against the tested bacteria, with MIC values below 12.61 mg/mL (Table 1). Throughout the development stages of TPL, the widest range of inhibitory effects against *S. aureus* and *L. monocytogenes* was observed, ranging from 3.13 to 12.61 mg/mL. This was followed by *P. vulgaris* and *S. Typhimurium*, with MIC values ranging from 3.13 to 6.31 mg/mL. *E. coli* exhibited an MIC range of 0.39 to 3.13 mg/mL, while *S. hemolytis-β* showed an MIC range of 0.2 to 0.39 mg/mL. These findings indicate that different stages of development have a significant impact on the ability of TPL to inhibit the growth of *S. aureus* and *L. monocytogenes*. In addition, TPL demonstrated particularly strong antibacterial properties against *S. hemolytis*-*β* and *E. coli*, with MIC values consistently below 3.13 mg/mL, highlighting its effectiveness against these strains compared to others.

The strongest activity against each bacterium was found in the early stages of TPL due to the lowest MIC values, which was in accordance with an early report [38]. As the leaves grew and developed, the antibacterial capacity against most bacteria decreased to a certain extent, eventually reaching a relatively stable MIC in the last stage. We observed that the antibacterial trend was consistent with the variations in TPC. A growing body of studies has demonstrated the effectiveness of polyphenolic compounds as natural antibacterial agents [41,42], suggesting that polyphenols of TPL may play critical roles in inhibiting bacterial growth. Considering the content and correlation coefficient, the antibacterial activity of TPL against *E. coli* is likely primarily attributed to the presence of benzoic acid, oxypaeoniflora, luteolin 7-glucoside, PGG, and gallic acid (Figure 7, Appendix A). In terms of the *L. monocytogenes* assay, antibacterial flavonoids, such as quercetin, (+)-catechin, kaempferol, isorhamnetin, and astragalin, were found to be responsible for the inhibitory activity of TPL (Figure 7, Appendix A). Although the correlation coefficients between other antibacterial assays and individual compounds were less than 0.4, TPL extracts exhibited strong inhibitory activities (Figure 7, Appendix A). This may suggest the presence of another effective inhibitory compound that was not quantified. Through Pearson correlation analysis, we also observed a close relationship between some species-specific monoterpenoid glycosides, such as benzoylpaeoniflorin and paeoniflorin, and *L. monocytogenes* and *E. coli* assays, even though their antibacterial activities have been rarely reported. The potential antibacterial mechanisms of those compounds warrant further investigation in future studies.

## 3. Materials and Methods

### 3.1. Chemicals and Reagents

Vanillin, hydrochloric acid, glacial acetic acid, ferric chloride, sulfuric acid, anhydrous sodium carbonate, sodium nitrite, sodium acetate, and other reagents were purchased from Bodi Chemical (Tianjin, China) and were analytically pure. HPLC-grade methanol and acetonitrile were purchased from Sigma-Aldrich (Saint Louis, MO, USA). The Folin–Ciocalteu regent, Trolox, 2, 4, 6-tris(2-pyridyl)-s-triazine (TPTZ), 1,1-diphenyl-2-picrylhydrazyl (DPPH), gallic acid, rutin, catechin, and 2, 2′-azinobis-3-ethylbenzothiazoline-6-sulfonic acid (ABTS) were acquired from Yuanye Bio-Technology (Shanghai, China).

### 3.2. Plant Materials

*P. ostii* ‘Feng Dan’ leaves were collected at different developmental stages from the Resource Garden of Northwest A&F University (34°15′ N, 108°03′ E, and Alt. 448 m) in 2021. These peonies were about 20 years old and had been acclimated in a resource nursery for over 5 years. The growth status of these trees was consistent and healthy. For each sample, six mono-leaves at the top of a plant were collected from six uniformly growing trees. Each sample was collected in triplicate every 10 days. The developmental period of leaves, from leaf unfolding to senescence, was divided into seventeen stages (S1–S17) (Figure 8, Appendix A).

### 3.3. Sample Preparation

After freeze-drying, the collected samples were crushed into a particle size of 60 mesh. Next, 0.2 g of the sample powder was subjected to ultrasonic extraction with 10 mL of methanol for 30 min. The supernatant of the mixture was then collected through centrifugation and used for TPC, TFC, LC-MS, and antioxidant assays. Prior to LC-MS analysis, the samples were filtered using a 0.22 µm filter. For the antibacterial assay, the freeze-dried leaves were extracted with methanol (*m*/*v* = 1:10) following the same conditions. The supernatant of the mixture was evaporated under vacuum conditions at 45 °C. The resulting dried extracts were accurately weighed and then redissolved for the antibacterial test.

### 3.4. Determination of Total Phenolic and Flavonoid Contents

The total phenolic content (TPC) and total flavonoid contents (TFC) of tree peony leaves at various growth stages were measured using the previously published protocol [43]. TPC and TFC were quantified and expressed as equivalents of gallic acid and rutin.

### 3.5. UPLC-ESI-QTOF-MS Analysis

Untargeted metabolomics analysis of *P. ostii* ‘Feng Dan’ leaves at different stages of development was conducted using a Shimadzu LC-30A UHPLC system equipped with an AB SCIEX Triple TOF 5600 plus mass spectrometer (Applied Biosystems, Foster City, CA, USA), following the methods described in our previous work [31]. Metabolites were separated through a Shimadzu Shim-pack XR-ODS column (100 mm × 2.0, 2.2 µm) (Kyoto, Japan) using a gradient elution with mobile phase A (0.1% formic acid) and B (acetonitrile) at a flow rate of 0.3 mL/min. The injection volume was 1 μL, and the column temperature was maintained at 35 °C. The gradient elution conditions were as follows: 16% B at 0.1 min, 24% B at 5 min, 55% B at 7 min, 65% B at 8 min, 98% B at 10 min, 98% B at 15 min, and 16% B at 18 min. Negative ionization mode was selected for QTOF MS analysis. Nitrogen was used as a nebulizer and auxiliary gas. The data were collected in information-dependent acquisition (IDA) mode, with a full scan range from *m*/*z* 100 to 1500. The parameters of the MS detector were set as follows: ion source gas 155 psi; ion source gas 255 psi; curtain gas 30 psi; source temperature 550 °C; ion spray voltage floating 4500 V; collision energy 35 eV; collision energy spread 15 eV; and declustering potential 80 eV. 

### 3.6. LC-ESI-QQQ-MS Analysis

Targeted metabolomics analysis was performed on *P. ostii* ‘Feng Dan’ leaves at different development stages using the AB SCIEX LC-ESI-QQQ-MS system (Applied Biosystems, Foster City, CA, USA). The method employed was consistent with our previous work [31]. Gradient elution was conducted on a Shimadzu AQ-C18 column (4.6 mm × 150 mm, 5 μm, Kyoto, Japan) with mobile phase A (0.1% formic acid) and B (acetonitrile) at a flow rate of 0.7 mL/min. The injection volume was 2 μL and the column temperature was set to 40 °C. The gradient elution conditions were as follows: 25% B at 0 min, 95% B at 6 min, 95% B at 9 min, 25% B at 9.1 min, and 25% B at 10 min. Negative ionization mode was selected in the QQQ MS analysis. Nitrogen was used as the GS1, GS2, and curtain gas. The MS detector’s common parameters were configured as follows: ion source temperature (T) 600 °C, ion spray voltage (IS) 4.5 kV, curtain gas 35 mL min^−1^, nebulizer gas 60 mL min^−1^, and heater gas 65 mL min^−1^. Targeted metabolites were detected in Multiple Reaction Monitoring (MRM) mode under optimized conditions for accurate quantification.

### 3.7. Determination of Antioxidant Activity

The DPPH free radical-scavenging activities of *P. ostii* ‘Feng Dan’ leaves at different stages were assessed following the methodology described by Taslimi et al., (2018) [44]. Briefly, 0.2 mL of sample extract was mixed with an equal volume of a 1 mM DPPH^•^ solution. The mixture was incubated for 45 min in the dark, and the absorbance was measured at 517 nm using a microplate reader (SP-MAX 2300A2, Shanpu, Shanghai, China). The antioxidant activity of the samples was expressed as Trolox equivalent (g TE/100 g) per 100 g dry sample.

The ABTS^+•^-scavenging activity of *P. ostii* ‘Feng Dan’ leaves at different stages was determined according to the method described by Cetin Cakmak, Kader [38]. Firstly, an ABTS radical cationic solution was prepared by mixing the same amount of 7 mM ABTS^+•^ with 2.45 mM potassium persulfate. The solution was then shaken uniformly and placed in the dark condition at room temperature for 12–16 h. Afterward, 2 mL of leaf extract and 2 mL ABTS^+•^ solutions were mixed and incubated for 6 min. The absorbance was finally measured at 734 nm. The antioxidant activity of each sample was expressed as 1 g Trolox equivalent (TE)/100 g dry weight (DW).

The ferric-reducing antioxidant power (FRAP) of *P. ostii* ‘Feng Dan’ leaves at different stages was determined following a published protocol [45]. The FRAP solution consists of 10 mM TPTZ, 40 mM HCl, 20 mM ferric chloride, and 0.3 M acetate buffer (10:1:1, *v*/*v*/*v*). The solution was thoroughly mixed and heated to 37 °C. A 0.1 mL sample solution was added to 0.9 mL of the reagent, and the mixture was incubated at room temperature for 30 min. Finally, the absorbance at 593 nm was measured. The antioxidant capacity of each sample was expressed as 1 gram of Trolox equivalent (TE) per 100 g of dry weight (g TE/100 g DW).

The assay of oxidative free radical absorption capacity (ORAC)of *P. ostii* ‘Feng Dan’ leaves at different stages was conducted using a modified method [46]. A total of 20 µL of sample was transferred to each well of a 96-well plate. The first control involved replacing the sample with 20 µL PBS buffer (75 mmol/L), followed by the sequential addition of 20 µL of PBS buffer and 20 µL of 70 nmol/L luciferase (dissolved in PBS buffer) to all the wells. After incubation at 37 °C for 5 min, 140 µL AAPH (12 mmol/L dissolved in PBS solution) was added to all samples, except for the second control well where the AAPH solution was replaced with an equal volume of methanol solution. The measurements were performed at an excitation wavelength of 485 nm and an emission wavelength of 538 nm, and readings were taken every 2 minutes for a total duration of 120 min using an all-wavelength multifunctional microplate reader (Infinite M200 Pro, Switzerland). The antioxidant results of the samples were expressed as Trolox equivalent (TE) per 100 g of dry weight (g TE/100 g DW).

### 3.8. Determination of Antibacterial Activity

The antibacterial activity of *P. ostii* ‘Feng Dan’ leaves at different stages was assessed against three Gram-positive bacteria (*Staphylococcus aureus* (ATCC25923), *Streptococcus hemolytis*-*β* (BNCC336670), and *Listeria monocytogenes* (ATCC19111)) and three Gram-negative bacteria (*Escherichia coli* (ATCC25922), *Proteus vulgaris* (ATCC6896), and *Salmonella enterica* (ATCC14028)). The minimum inhibitory concentration (MIC) was employed to evaluate the antibacterial activity, following the established method [47]. In brief, the 200 mg/mL samples were double-diluted using a 2% DMSO-PBS solution [27]. Next, 100 µL of the sample extracts at different concentrations were added to sterile 96-microwell plates, followed by the addition of 100 µL of the corresponding bacterial suspension. The blank control well contained 200 µL of 2% DMSO-PBS solution, while the self-growth control well contained 100 µL of 2% DMSO-PBS solution and 100 µL of bacterial solution. After incubation at 37 °C for 18 h, 50 µL of TTC (2,3,5-triphenyl tetrazolium chloride) was added to all wells. The MIC value was determined as the concentration of the extraction solution in the previous well showing obvious discoloration after incubation at 37 °C for 6 h.

### 3.9. Statistical Analysis

All values are reported as mean ± standard deviation. Statistical analysis was performed using SPSS 19.0 software. The raw data collected by UPLC-QTOF-MS were processed using the open software package MZmine 2 to obtain the peak list of the 1353 × 55 matrix [48]. The matrix was then imported into SIMCA-P 14.1 for principal component analysis (PCA) and orthogonal partial least-squares discriminant analysis (OPLS-DA). A histogram drawing was generated using GraphPad Prism 9.0.0 software. Principal component analysis, cluster analysis, and graph drawing were performed using SIMCA 14.1 software. Heat maps were created using TBtools v.1.108 software. Kyoto Encyclopedia of Genes and Genomes (KEGG) enrichment analysis diagrams were drawn using the OmicStudio online tool (https://www.omicstudio.cn/tool) accessed on 1 January 2022. The MS/MS molecular network was constructed using the GNPS (Global Natural Products Social Molecular Networking) platform, and data visualization was carried out using Cytoscape 3.8.2 software. 

## 4. Conclusions

In the present study, the phytochemicals present in *P. ostii* ‘Feng Dan’ leaves at different stages of development were systematically investigated using an integrative metabolomics approach. A total of 321 metabolites were comprehensively annotated based on public or in-house MS databases, many of which were previously unreported in tree peony leaves. The transition of peonies from vegetative to reproductive growth may be the turning point resulting in the trend of metabolite variation. Significant variations in metabolites were observed throughout leaf development, particularly for dominant compounds such as albiflorin, paeoniflorin, pentagalloylglucose, luteolin 7-glucoside, and benzoylpaeoniflorin. These variations can be attributed to the changes in antioxidant and antibacterial activities. The extract of TPL exhibited the strongest inhibitory potential against *S. hemolytis-β* among all tested bacteria. These findings suggest that the stages of S1–S2 may be the most promising harvesting time when considering both phytochemicals and bioactivities. This work provides a theoretical basis for the comprehensive exploitation and utilization of *P. ostii* ‘Feng Dan’ leaves for functional tea or pharmaceutical purposes. However, considering the influences of environmental factors, the pattern of metabolite variation still needs to be investigated in the next growing season.

## Figures and Tables

**Figure 1 ijms-25-01059-f001:**
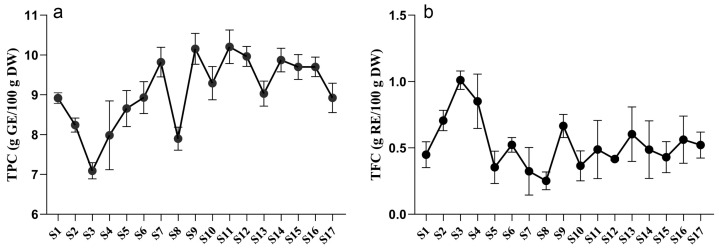
Dynamic variations of TPCs (**a**) and TFCs (**b**) in *P. ostii* ‘Feng Dan’ leaves during the whole developmental stages. TPC, total phenolic content. TFC, total flavonoid content.

**Figure 2 ijms-25-01059-f002:**
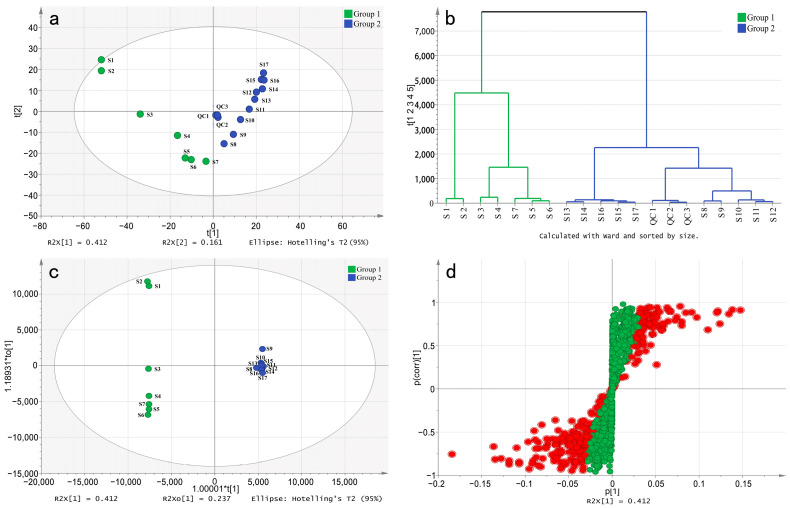
Multivariate statistical analysis of *P. ostii* ‘Feng Dan’ leaves at different stages. (**a**), the PCA score scatter plot; (**b**), the HCA plot of the PCA; (**c**), the OPLS-DA score scatter plot; (**d**), the S-plot of the OPLS-DA. Red dot, VIP > 1; green dot, VIP < 1.

**Figure 3 ijms-25-01059-f003:**
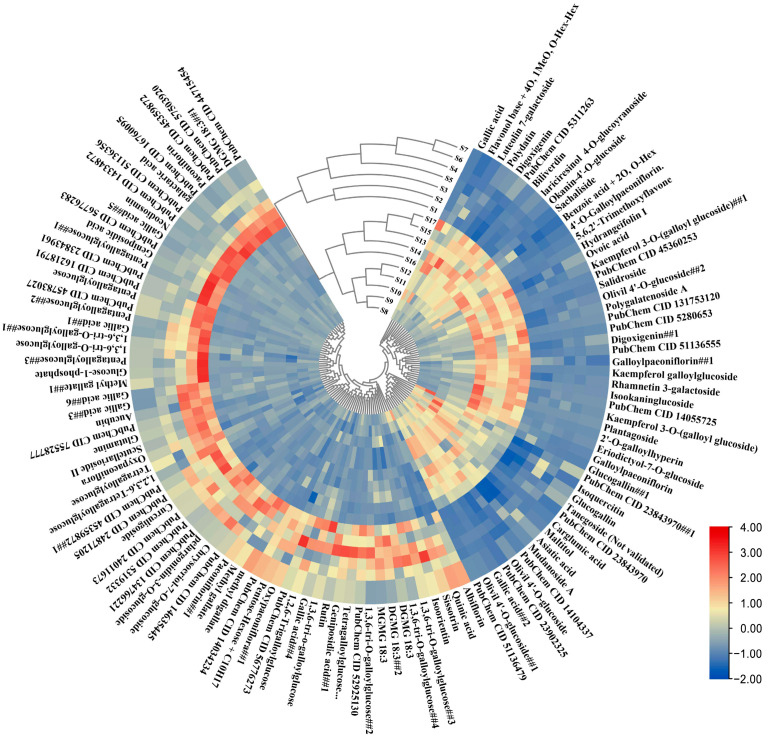
Heat map of 117 important differential metabolites of *P. ostii* ‘Feng Dan’ leaves at different stages.

**Figure 4 ijms-25-01059-f004:**
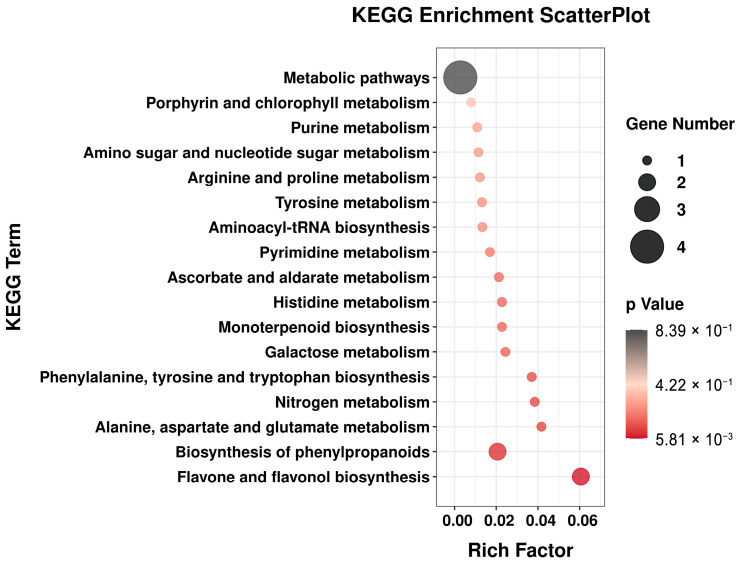
Differential metabolite KEGG enrichment scatter plot of *P. ostii* ‘Feng Dan’ leaves between Group 1 and Group 2.

**Figure 5 ijms-25-01059-f005:**
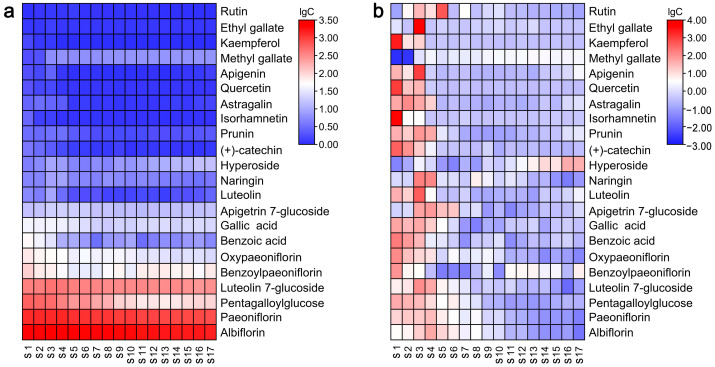
Heatmap of targeted metabolites in *P. ostii* ‘Feng Dan’ leaves at different stages. (**a**), plotted with all target metabolites; (**b**), plotted with each target metabolite in a row independently.

**Figure 6 ijms-25-01059-f006:**
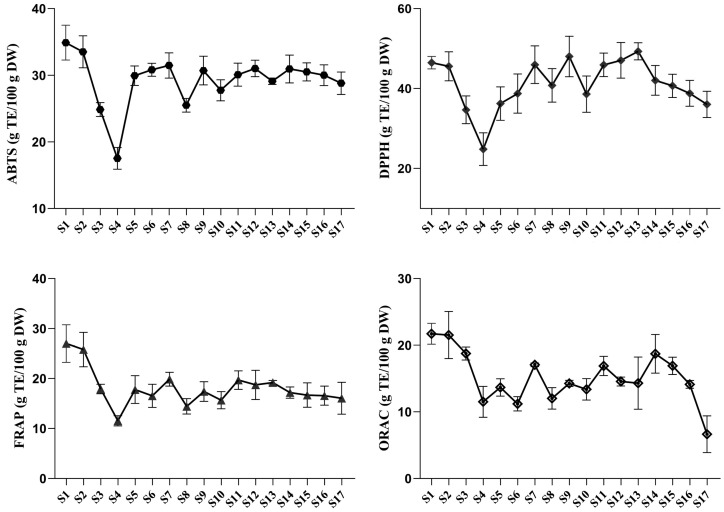
The antioxidant activities of *P. ostii* ‘Feng Dan’ leaves at different stages in vitro. Spot and error bar represent mean value and standard deviation (*n* = 3).

**Figure 7 ijms-25-01059-f007:**
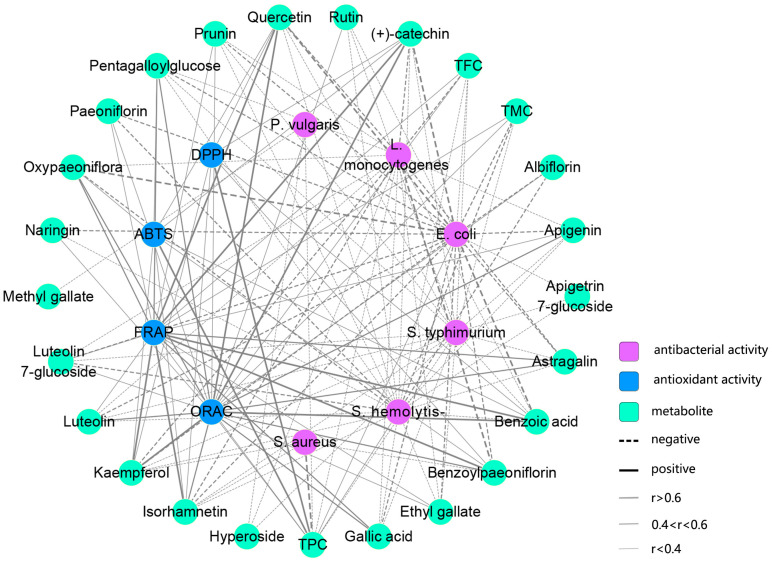
Correlation network between antioxidant activity and bioactive constituents. TMC, total metabolite content of targeted metabolites.

**Figure 8 ijms-25-01059-f008:**
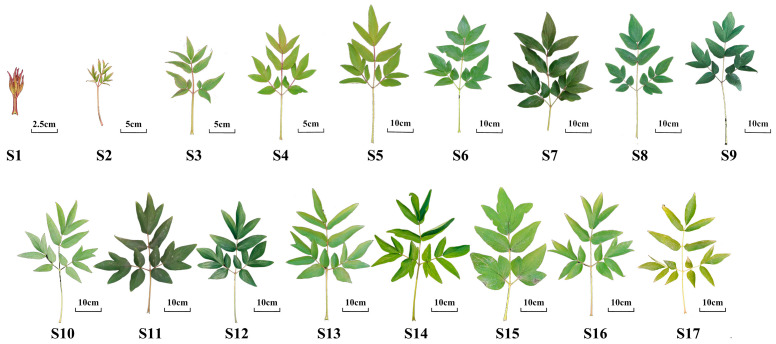
Phenotypes of *P. ostii* ‘Feng Dan’ leaves during all developmental stages.

**Table 1 ijms-25-01059-t001:** MICs (mg/mL) of *P. ostii* ‘Feng Dan’ leaves at different stages.

Stage	Gram-Negative Bacteria			Gram-Positive Bacteria		
	*E. coli*	*P. vulgaris*	*S. typhimurium*	*S. aureus*	*S. hemolytis-β*	*L. monocytogenes*
S1	0.39	6.25	3.13	3.13	0.39	3.13
S2	0.39	6.25	3.13	3.13	0.39	6.25
S3	0.39	6.25	3.13	12.50	0.39	6.25
S4	1.56	6.25	3.13	12.50	0.39	6.25
S5	1.56	6.25	3.13	6.25	0.39	12.50
S6	3.13	3.13	3.13	6.25	0.20	12.50
S7	1.56	3.13	1.56	3.13	0.39	6.25
S8	1.56	6.25	6.25	6.25	0.39	12.50
S9	1.56	6.25	6.25	6.25	0.39	6.25
S10	1.56	6.25	3.13	3.13	0.39	12.50
S11	1.56	3.13	3.13	3.13	0.39	6.25
S12	1.56	6.25	3.13	6.25	0.78	6.25
S13	1.56	6.25	3.13	3.13	0.39	6.25
S14	1.56	3.13	3.13	3.13	0.78	6.25
S15	1.56	6.25	3.13	3.13	0.39	6.25
S16	3.13	6.26	3.13	6.26	0.78	6.26
S17	3.14	3.14	3.14	6.28	0.39	6.28

## Data Availability

All relevant data are included in the manuscript and Appendix A.

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
