# Peer review of "Integrated Metabolomics Approach Reveals the Dynamic Variations of Metabolites and Bioactivities in Paeonia ostii ‘Feng Dan’ Leaves during Development"

_ijms, 2024, doi:10.3390/ijms25021059_

Round 1

Reviewer 1 Report

Comments and Suggestions for Authors

Zhangzhen Bai et al. describe a metabolomics-based study on Paeonia ostii 'Feng Dan' leaves during development. In addition, antioxidant and antibacterial were further evaluated. The manuscript can be considered interesting and has relevant elements for readers. However, some substantial shortcomings should be addressed before further consideration:

1.     The format of the journal template should be carefully followed.

2.     Title: Why Integrated? This study did not combine different analytical platforms or distinct omics to be considered integrated. So, this term must be removed from the title to avoid confusing readers' interpretations. The integration needs to support a comprehensive analysis from different perspectives. Be consistent throughout the manuscript.

3.     Line 23-24: The authors should specify the most promising harvesting times for producing/accumulating interesting metabolites.

4.     Line 32: The citation format is not correctly followed.

5.     Lines 86-90: The authors must specify more details about plant material collection. For instance, the age, status, strata, sanitary status, environmental conditions, season, etc, must be provided to ensure reproducibility. In addition, were the leaf samples collected every ten days from the same tree? If yes, are the authors sure this abscision did not affect the metabolic response due to mechanical damage? This fact should be considered in the R&D section.

6.     Line 92: In the Figure 1 caption, define what the letter "Y" means in the groups Y1-toY17. Are they related to specific time frames associated with developmental stages? Specify it. A Figure must be self-explanatory. Be consistent throughout the manuscript since several figura captions do not adequately explain the Figures' content.

7.     Line 95: Did the authors consider that ultrasound-assisted extraction might affect metabolites to produce artifacts? Did the authors perform a previous assay to asses the integrity after extraction using ultrasound? This explanation must be provided to readers.

8.     Lines 105 and 109: The column, mobile phase, and gradient elution for the chromatographic conditions and MS parameters must be informed in this study for readers.

9.     Line 208: The main limitation and concern of this study is the compound identification, which seems incomplete. Checking the supplementary material 2 (Table S2), I found some flaws in the compound identification. For instance, compounds numbered 37, 61, 64, and 87, with different retention times (i.e., 1.22, 2.65, 2.77, and 3.71 min), have the same accurate mass (m/z 147.0296) and the same identification (i.e., 2-hydroxyglutaric acid), which has no sense since if they have different retention times must be related to different compounds (at least isomers). However, the authors did not report it or the confidence level for their compound identification. Several other examples are found in this table s2 (for instance, compounds 278, 120, 132, 183, 188, 202, 94, and 102 linked to methyl gallate, or compounds 101, 20, 26, 80, 42, 34, 49, and 53 linked to gallic acid). This shortcoming enormously limits this study and generates doubts about the quality of the result since the main findings and discussion are based on this identification. In fact, the total number is not related to 321 metabolites since several features were related to the same metabolite. The authors must detail how the compound identification per metabolite was achieved since the mere comparison of MS data with literature or databases is insufficient to ensure proper identification. Additionally, the authors must add a level of confidence for communicating the identification of compounds detected by HRMS and/or MS/MS.

10.  Fig 4. Details for compound identification must be provided.

11.  Line 226: The quality parameters and model type for OPLS-DA are missing and must be informed.

12.  The manuscript is highly descriptive, and the discussion is inadequate. In other words, the discussion should be improved since the manuscript describes results and includes some introductory ideas; however, a good comparative discussion with previous studies and theories is missing. In fact, the R&D section only compares earlier studies from the same authors, but comparisons with other studies/theories are missing. For instance, but not the only issue, sections 3.5 and 3.6 did not compare results and only described the study's outcome laconically.

13.  Conclusions summarize results, and conceptual findings from the mechanistic point of view must be provided for readers. Therefore, this section can be improved to deliver a clear, conclusive message from the study results.

Comments on the Quality of English Language

The authors must revise the grammar and style of the manuscript since some passages are challenging to follow. Detailed scrutiny via language editing is recommended.

Reviewer 2 Report

Comments and Suggestions for Authors

The article titled "Integrated Metabolomics Approach Reveals the Dynamic Variations of Metabolites and Bioactivities in Paeonia ostii 'Feng Dan' Leaves during Development" fits the scope of the journal. The authors examined differences in the content of metabolites and biological activity of tree peony leaf extracts during the growing season. This is a very interesting study that provides important information for the herbal industry. The article is well written, the research is planned logically and purposefully. I appreciate the amount of work the authors put into the research. The article can be published after minor corrections:

The results and discussion part is a bit too general in my opinion. There is no detailed comparison of the results of the work with the research of other authors. I would like to ask for a more detailed comparison with the list of individual metabolites and their content obtained in other studies.

The authors obtained very surprising results in their study of TPC values. Could the authors try to explain why TPC suddenly drops at stage Y8? In the case of TFC, the overall trend is rather decreasing, but also at stage Y8 it is the lowest, then increases slightly. Could the authors explain the reason? Were any weather anomalies observed during this time? Maybe the plants were then subjected to antifungal sprays or other breeding activities?

The above implies the need to conduct the research again in the next growing season, especially since the authors propose the best date for harvesting the raw material. I propose to place such a proposal in the Conclusion section. This section should include proposals for further research in the discussed area.

I ask the authors to correct the references in accordance with the journal's guidelines. References of Journal Articles should be described as follows:
1. Author 1, A.B.; Author 2, C.D. Title of the article.
Abbreviated Journal Name Year, Volume, page range.

Reviewer 3 Report

Comments and Suggestions for Authors

The manuscript ijms-2789956 titled “Integrated Metabolomics Approach Reveals the Dynamic Variations of Metabolites and Bioactivities in Paeonia ostii ‘Feng Dan’ Leaves during Development” is the attempt follow the transformation of polyphenolic metabolites in plant leaved during the development of the selected plant.
Generally, it is good experimental work that involves chromatographic measurements and statistical treatment of the obtained data. Despite the fact that the task is complicated and the obtained results and interpretation are objects of discussion, I believe that the presented experimental data will be interesting for readers of IJMS and other experts in the field.

Comments:
1. Please clearly describe what is presented in Figure 2. I suggest the correction of the figure caption with the addition of a) and b). What is the title of the x-axis, Sample? Sample names are not mathematical values. May it be better to substitute the Sample name with Time?
Same for Figure 7.
2. Please make sure that all abbreviations are described at the first mention in the text. A description is missing for some of them. For example, KEGG lines 171, 247, and 250.
3. Formatting of the manuscript is needed.
4. Please increase the text size in Figures 3 and 8.

Comments on the Quality of English Language

 Minor editing of English language required.

Reviewer 4 Report

Comments and Suggestions for Authors

The manuscript with the title “Integrated Metabolomics Approach Reveals the Dynamic Variations of Metabolites and Bioactivities in Paeonia ostii ‘Feng Dan’ Leaves during Development”, presents the findings of metabolite variations during leaf development of an edible and ornamental peony cultivar. The manuscript has good potential to be attractive to readers.

Abstract provides a relevant summary of the study.

Introduction is well-written and in tune with the study. I suggest authors to add some brief information on botany, general peony phenology, and the basic requirements for environmental factors on this species.  

Comment on chapter 2.2. Plant materials

Line 89 “from leaf germination to yellow”, did authors mean from leaf unfolding until senescence? Please be specific on the phenophase. The leaf germination is not suitable term here.
Each stage has to be named specifically (Table S1 with the phenology should be here in the material and method section!), the leaf phenology and metabolite content fluctuations are related to the general plant phenology ideed. E.g. the leaf collected in a vegetative stage of a plant is expected to have a different phytochemistry than the leaf collected when the plant is at flowering or fruiting stage. Because phenology is in close relationship with light, temperature and precipitations – the environmental conditions that plants enjoyed should have been detailed as well. E.g. brief description of the climate/seasons, or a graph with temperature and precipitations during the interval of study. Etc.

Comment on chapter 2.8.

The samples for the antibacterial activity was represented by freezer-dried leaves also?

Results

Results are well detailed, but findings for Y-1 to Y-17 are not explained in the text in connection with the growth stages (those stages from Table S1). For example, what was the main trend found between vegetative versus generative stages in terms of leaf metabolites and matabolom? After reading the paper this is not very clear. The most important findings have to be well highlighted, it is what readers are looking for.

I hope my comments can help authors to improve their manuscript.

Best regards.

Comments on the Quality of English Language

moderate improvements of English style and grammar are needed.

Round 2

Reviewer 1 Report

Comments and Suggestions for Authors

The authors have adequately addressed my comments, resulting in a significant improvement in the quality and content of the manuscript. Additionally, my main concern on compound identification has been appropriately addressed. The terminology and scope for HRMS and MS/MS analyses within a metabolomics approach now allow a proper scope for accurately communicating metabolite identity with correct confidence using these analytical platforms. Consequently, my initial perception has changed since the authors performed the necessary revisions, and the manuscript can be accepted in its current form.

Reviewer 4 Report

Comments and Suggestions for Authors

Dear authors,

the manuscript was improved for clarity and readability. Main issues were explained and answered in your responses.

Best regards.

Comments on the Quality of English Language

minor English style improvements are recommended.